# A Systematic Review on Metabolomics Analysis in Hearing Impairment: Is It a Possible Tool in Understanding Auditory Pathologies?

**DOI:** 10.3390/ijms242015188

**Published:** 2023-10-14

**Authors:** Rita Malesci, Martina Lombardi, Vera Abenante, Federica Fratestefano, Valeria Del Vecchio, Anna Rita Fetoni, Jacopo Troisi

**Affiliations:** 1Department of Neuroscience, Reproductive Sciences and Dentistry (Audiology and Vestibology Service), University of Naples Federico II, 80138 Napoli, Italy; valeria.delvecchio@unina.it (V.D.V.); annarita.fetoni@unina.it (A.R.F.); 2Theoreo srl, Spin off Company of the University of Salerno, Via Degli Ulivi 3, 84090 Montecorvino Pugliano, Italy; abenante@theoreosrl.com (V.A.); fratestefano52@gmail.com (F.F.); troisi@theoreosrl.com (J.T.); 3Department of Chemistry and Biology “A. Zambelli”, University of Salerno, 84084 Fisciano, Italy; 4European Institute of Metabolomics (EIM) Foundation ETS, G. Puccini, 2, 84081 Baronissi, Italy; 5Department of Medicine, Surgery and Dentistry “Scuola Medica Salernitana”, University of Salerno, 84081 Baronissi, Italy

**Keywords:** hearing loss, deafness, ear, metabolomics, metabolites, audiology, biomarkers

## Abstract

With more than 466 million people affected, hearing loss represents the most common sensory pathology worldwide. Despite its widespread occurrence, much remains to be explored, particularly concerning the intricate pathogenic mechanisms underlying its diverse phenotypes. In this context, metabolomics emerges as a promising approach. Indeed, lying downstream from molecular biology’s central dogma, the metabolome reflects both genetic traits and environmental influences. Furthermore, its dynamic nature facilitates well-defined changes during disease states, making metabolomic analysis a unique lens into the mechanisms underpinning various hearing impairment forms. Hence, these investigations may pave the way for improved diagnostic strategies, personalized interventions and targeted treatments, ultimately enhancing the clinical management of affected individuals. In this comprehensive review, we discuss findings from 20 original articles, including human and animal studies. Existing literature highlights specific metabolic changes associated with hearing loss and ototoxicity of certain compounds. Nevertheless, numerous critical issues have emerged from the study of the current state of the art, with the lack of standardization of methods, significant heterogeneity in the studies and often small sample sizes being the main limiting factors for the reliability of these findings. Therefore, these results should serve as a stepping stone for future research aimed at addressing the aforementioned challenges.

## 1. Introduction

Hearing loss (HL) is a prevalent sensory disorder with far-reaching consequences on an individual’s life, encompassing social, functional, and psychological aspects [1]. Deprivation of auditory abilities can hinder access to spoken communication, impeding the development of language skills and executive functioning in children [2] and contributing to the risk of dementia and cognitive decline in older age groups [3].

HL affects 466 million people worldwide and it is estimated that in 2050 over 700 million people will need rehabilitation support for HL (https://www.who.int/, accessed on 27 June 2023).

Sensorineural hearing loss (SNHL) stands as the most prevalent form of hearing impairment, involving various pathologies within the inner ear and auditory nerve. The primary mechanism behind SNHL lies in the damage or loss of sensory receptor cells in the cochlea and the neurons responsible for relaying auditory information to the central auditory system. As a result, SNHL presents as a chronic condition for which rehabilitative, rather than curative, treatment option exist [4].

SNHL can be congenital or acquired. The most common causes of permanent congenital sensorineural are cytomegalovirus infection, structural abnormalities of the temporal bones, and genetic causes [5]. The principle causes of acquired SNHL are presbycusis, acoustic trauma, ototoxic drugs, sudden HL, and infectious conditions [6,7]. A significant challenge in managing SNHL lies in the limited understanding of its pathogenetic mechanisms. Although several models have been proposed to explain the molecular pathways leading to hair cell death and damage to spiral ganglion neurons, common pathways have been observed across different damage etiologies, including the accumulation of reactive oxygen species (ROS), activation of MAPK signaling, and apoptosis pathways [8].

Contemporary research endeavors are dedicated to identifying predictive elements with diagnostic and prognostic value, aimed at improving the management and understanding the evolution of SNHL. Indeed, the integration of precision medicine principles into translational HL research and clinical practice holds the potential to offer personalized medical care, addressing the diverse etiologies of HL with greater precision and efficacy.

The past two decades have witnessed a veritable explosion of new omics technologies, which have contrasted the reductionist approach, typical of traditional biology, with a broader holistic one [9]. This shift in perspective marked a real revolution in the biomedical field, especially in dealing with complex problems, such as HL, characterized by multifactorality and high interindividual variability [10]. 

Metabolomics is an emerging omics science, which studies the set of metabolites, i.e., the metabolome, contained within a biological system. As the metabolome lies downstream from the central dogma of molecular biology, it is able to reflect both the genetic characteristics and the environmental influences. Moreover, due to its dynamic nature, the metabolome undergoes rapid and well-defined changes in case of disease [11,12]. As a result, metabolomic analysis opens a unique window on the mechanisms underlying the different phenotypes.

There are two possible approaches in metabolomics, targeted and untargeted. The former is aimed at quantifying a set of specific metabolites, and is generally chosen to test a hypothesis. The second, on the other hand, allows the entire metabolome to be explored without starting from preexisting hypotheses, but rather allowing new ones to be formulated. Both of these approaches have already provided important results in numerous fields of medicine, ranging from the study of cancer [13,14,15] to fetal malformations [16,17,18], psychiatric, neurodevelopmental and neurodegenerative disorders [19,20,21,22,23] as well as chronic and autoimmune diseases [24,25] and internal medicine pathologies [26,27,28], just to name a few. 

In recent years, various research groups have tried to apply metabolomics investigations to the study of HL, using both the targeted and the untargeted approaches. The purpose of this review is to collect and analyze the current state of the art about metabolome changes in HL patients and to identify the main future perspectives in the field. 

## 2. Methods

The present review was written in accordance with the PRISMA (Preferred Reporting Items for Systematic Review and Meta-Analysis recommendations) methodology [29]. A systematic literature search was carried out using PubMed as database in May 2023. Publications were obtained using the following search query: (metabolomics OR metabonomics OR metabolites) AND (hearing loss OR deafness). Three authors independently selected papers, and in case of disagreement final decision was entrusted to a senior author. As a result, a total of 251 peer-reviewed articles were identified and screened. After careful examination, articles were excluded if they (a) were not in English, (b) were case reports, (c) did not have full text available, (d) did not include metabolomics analyses. A total of 20 original articles were included in the qualitative synthesis. The flow diagram of the study selection process is shown in Figure 1.

## 3. Results 

### 3.1. Metabolomic Profiling in Humans with Hearing Loss 

The main characteristics of the studies carried out in humans are summarized in Table 1. Of the 13 studies included, 8 were performed on urine samples [30,31,32,33,34,35,36,37]; 1 on perilymph [38], 2 on plasma [39,40] and 2 on serum [41,42]. The most commonly used technique for metabolomic profiling was liquid chromatography coupled with mass spectrometry (LC-MS); while only one study was performed by Nuclear Magnetic Resonance (NMR) and two by Gas Chromatography-Mass Spectrometry (GC-MS). Appendix A reports a brief description of these analytical technologies with their advantages and disadvantages for both targeted and untargeted metabolomics.

#### 3.1.1. Metabolomic Analysis on Perylimph Samples

Metabolomic analyses on perilymph samples from individuals with SNHL were performed by Trinh et al. [38], who investigated the relationship between metabolomic profiles and the duration of HL in cochlear implanted patients. This study revealed significant differences in the metabolomic profiles of patients with more and less than 12 years of HL, with N-acetylneuraminate showing a strong positive correlation with the duration of HL. 

#### 3.1.2. Metabolomic Analysis on Blood-Derived Samples

Wang et al. [42] investigated sudden sensorineural hearing loss (SSNHL) and its metabolic implications. The research included 20 SSNHL patients and 20 healthy controls (HCs), with serum samples collected for analysis. Metabolites were detected using LC-MS, and distinct metabolic profiles were observed in SSNHL patients compared to HCs, with a significant alteration in fatty acids metabolism. The patients were then categorized into recovery and non-recovery groups, and 12 distinctive metabolites were observed between the two groups, with significant positive associations between serum N4-Acetylcytidine, sphingosine, and nonadecanoic acid levels with hearing recovery in SSNHL patients. Metabolomic investigations were also used to assess the correlation between age ARHL and other pathological conditions. Using a publicly available dataset of the Alzheimer’s Disease Neuroimaging Initiative (ADNI), Llano et al. [41] investigated the serum lipidomic profiles in Alzheimer’s Disease (AD) subjects with and without HL. An interesting result was obtained for phosphatidylcholines, known to be one of the most important phospholipids of cell membrane, that showed a decreased abundance in AD patients with HL. A recent study [39] investigated the plasma metabolomic profile of individuals suffering from noise-induced hearing loss (NIHL). As a result, a total of seven pathways emerged as being involved, including glycerophospholipid metabolism, glycosylphosphatidylinositol (GPI) anchor biosynthesis, autophagy pathway, choline metabolism, the alpha-linolenic acid metabolism and linoleic acid metabolism, and retrograde endocannabinoid pathway. To further investigate how noise exposure affects the autophagy pathway, as part of this study the authors assessed the expression of three autophagy-related genes, namely PI3K, AKT and ATG5, which were found to be significantly down-regulated in NIHL patients compared to controls. Zhang et al. [40] studied noise-induced metabolic dysregulation by analysing a total of 90 samples collected from 60 noise exposed individuals, half of whom suffering from NIHL, and 30 HC. A total of 6 pathways resulted to be associated with both NIHL and noise exposure, including retrograde endocannabinoid signaling, sphingolipid signaling pathway, vitamin digestion and absorption, Fc gamma R-mediated phagocytosis, phospholipase D signaling pathway, and central carbon metabolism in cancer. Among the 7 metabolites emerged as crucial in differentiating NIHL patients from the other two groups, Pro-Trp dipeptide, adenine, and dimethylglycine exhibited a progressive decline pattern, starting from the control group to the non-NIHL group, and reaching the lowest levels in the NIHL group. These altered levels suggest that these metabolites may play significant roles in the development of noise-induced disorders. In addition, the researchers conducted cochlear gene expression comparisons between mice susceptible and resistant to NIHL using the GSE8342 dataset from Gene Expression Omnibus (GEO). The analysis revealed a strong association between immune response and cell death-related processes and noise exposure, suggesting their crucial involvement in noise-induced disorders.

#### 3.1.3. Metabolomic Analysis on Urine Samples

Carta et al. [34] focused on individuals with idiopathic sudden-onset sensorineural hearing loss (ISSNHL) who underwent a treatment regimen involving steroids, heparin, and hyperbaric oxygen therapy. Through NMR-based urine metabolomic profiling, notable distinctions emerged between subjects who positively responded to steroid therapy and those who did not. Specifically, the non-responders exhibited elevated levels of β-alanine, trimethylamine N-oxide (TMAO), and 3-hydroxybutyrate, while citrate and creatinine showed an opposite pattern.

In another study [35], through GC-MS analysis on urine samples from individuals with presbycusis and normal hearing matched controls, a total of 23 metabolites were found to be differently expressed. KEGG (Kyoto Encyclopedia of Genes and Genomes) pathway analysis revealed that these metabolites were mainly involved in the metabolism of glutathione, amino acids, glucose, and related to GABA and NMDA receptors activity. The findings also showed a notable disparity in the accumulation of kidney deficiency points between the presbycusis group and elderly individuals with normal hearing. This is a scoring system for evaluating symptoms associated with kidney deficiency, as formulated in accordance with the ‘Standards of Practice for Assessing Grading Factors of Kidney Deficiency Syndrome Differentiation’ as revised by the ‘Standards of Reference for Traditional Chinese Medicine Deficiency Syndrome’, under the auspices of the Chinese Integrated Traditional and Western Medicine Deficiency Syndrome and Geriatrics Research Professional Committee. Thus, the authors posit that these identified metabolic pathways could potentially serve as the underlying basis for the connection between presbycusis and kidney deficiency, as postulated in traditional Chinese medicine. Moreover, in a retrospective study by Pudrith and Dudley [36], the urinary levels of volatile organic compounds (VOCs) were used to investigate the implication of oxidative stress in acquired sensorineural hearing loss (ASNHL), considering that reliable biomarkers are lacking [43]. The study revealed a significant increase of around 3 to 4 dB in high-frequency pure-tone thresholds among individuals in the upper quartile groups of five specific metabolites. These metabolites were identified as glutathione-dependent mercapturic acids, originating from parent compounds such as acrylonitrile, 1,3-butadiene, styrene, acrylamide, and N,N-dimethylformamide. Notably, these associations were evident only in individuals who reported no exposure to noise.

Other studies have focused on the correlation between exposure to specific chemicals and auditory system function. Two studies investigated the implications of polycyclic aromatic hydrocarbons (PAHs) in HL [31,33]. PAHs are environmental pollutants which are taken up by humans through inhalation, ingestion or even dermal exposure and have been associated with numerous harmful effects on human health, including the induction of oxidative stress [44]. Chou et al. found a dose-response relationship between exposure to PAHs and increased hearing thresholds in adults, emphasizing the potential role of these metabolites in causing damage to the cochlea, which may be a result of the inflammatory response they induce [33]. Following the same line of research, Li et al. [31] investigated the association between PAHs and HL as well as the potential role of systemic inflammation as a key mediator of this correlation in a large cohort of adults and adolescents. In this study, a significant association between the urinary levels of PAHs and HL was confirmed in both groups, hence suggesting that monitoring these levels may be a useful strategy to prevent HL. Interestingly, C-reactive protein, which is a valuable marker of inflammatory response, was found to have no mediating effects in the relationship between PAHs and HL. Other compounds of growing concern that have already been linked to neurotoxicity phenomena are pyrethroids, commonly used as insecticides. In a study by Xu et al. [30], urinary levels of 3-phenoxybenzoic acid (3-PBA), a pyrethroid metabolite, were evaluated to assess pyrethroid exposure in a wide cohort of adolescents. The study revealed a strong association between the exposure to these compounds and an increase in hearing thresholds. Similarly, Long & Tang [37] further investigated the ototoxicity induced by pyrethroid exposure in a large cohort of adults. In this case, an association between 3-PBA levels and both low and high frequencies hearing thresholds was described in subjects aged 20–39 years. In the pursuit of understanding the potential harmfulness of specific environmental substances, a study investigated the impact of caffeine exposure on hearing thresholds. However, no association between caffeine metabolites and hearing function was observed in US adults [32]. 

### 3.2. Metabolomics Investigations on Animal Models of Hearing Loss

Table 2 shows the key features of the included studies on animal models. Of them, 3 were performed on guinea pigs [45,46,47], 1 on rats [48], 2 on mice [49,50] and 1 on sheep [51]. Metabolomic profiling was conducted using LC-MS techniques in four studies [45,47,49,50], while in the others GC-MS was applied [46,48,51].

#### 3.2.1. Metabolomic Analysis on Tissues

In a study on mice by Ji et al. [49], a targeted metabolomic analysis for 220 metabolites was performed to investigate the effects of noise exposure on inner ear. Inner ear samples were collected immediately after noise exposure. A total of 40 metabolites were found to be affected by noise, with metabolic alterations depending on the intensity and duration of noise exposure. The main metabolic pathways altered by noise were purine metabolism and alanine, aspartate and glutamate metabolism, which were found to be upregulated, while phenylalanine metabolism and phenylalanine, tyrosine and tryptophan biosynthesis were downregulated. In a study carried out by Miao et al. [50] involving mice, when analyzing cochlea tissues, 17 metabolites demonstrated statistical significance in differentiating between noise-exposed mice and control subjects, with a total of 9 metabolic pathways involved. Among these metabolites, only three, namely spermidine, 3-hydroxybutyric acid, and orotic acid, showed higher levels in the noise-exposed group as compared to the controls. Interestingly, when examining rat brains exposed to acoustic trauma, the metabolomic analyses showed diverse metabolic patterns among different regions, yet regions with similar functions displayed analogous metabolite compositions [48]. Notably, in certain brain areas, such as the auditory cortex, 17 crucial metabolites were identified, distinguishing between control and acoustic trauma-exposed animals. These metabolites predominantly involved amino acid metabolism, specifically affecting the alanine, aspartate, glutamate, arginine, proline, and purine metabolic pathways.

#### 3.2.2. Metabolomic Analysis on Biofluids

Fujita et al. [46] performed two distinct experiments on guinea pigs. The first involved a comparison of metabolite abundance in the inner ear fluid and plasma under normal conditions, showing differences in the levels of 15 metabolites. Then, they investigated the metabolomic alterations after exposing the guinea pigs to loud noise. As a result, a total of ten metabolites showed altered concentrations after noise exposure, namely 3-hydroxy-butyrate, glycerol, fumaric acid, galactosamine, pyruvate + oxaloacetic acid, phosphate, meso-erythritol, citric acid + isocitric acid, mannose, and inositol. The characterization of “inner ear fluid” as per the authors denotes a composite fluid resulting from the amalgamation of both endolymphatic and perilymphatic fluids. This definition aligns with the methodology employed, wherein the authors meticulously extracted this fluid subsequent to the removal of temporal bones, employing a dissecting microscope. This process involved delicately accessing the lymphatic fluid within the cochlea via the round and oval windows. While it is probable that the bulk of the fluid extracted corresponded to perilymphatic fluid, it is conceivable that endolymphatic fluid might have also been encompassed within the collection. 

The metabolomic alterations following acoustic trauma were also investigated on perilymph samples extracted from sheep in a study by Boullard et al. [51]. In this context, the sheep served as their own reference for comparison. Indeed, for each sheep only one ear was subjected to acoustic trauma, representing the NIHL model, while the other was used as control. As a consequence of noise exposure, 5 metabolic pathways resulted affected, namely phenylalanine/tyrosine/tryptophan metabolism, β-alanine metabolism, pantothenate and CoA biosynthesis, pyrimidine metabolism, and amino and nucleotide sugar metabolism.

Moreover, an untargeted metabolomic analysis was performed on perilymph samples collected from guinea pigs exposed to a heavy impulse noise [45]. As part of this study, the potential otoprotective role of Hydrogen gas (H_2_) inhalation was also investigated. Animals were divided into four subgroups: controls; animals exposed to noise only; exposed to noise and treated with H_2_; only treated with H_2_. Guinea pigs exposed to noise and receiving H_2_ treatment were found to have a reduced HL compared to those in the noise only group. Metabolomic analysis confirmed the otoprotective role of this gas by showing that the perilymphatic metabolome of H_2_-treated animals exposed to noise was more similar to that of the controls and H_2_ only groups than to the noise only group. In particular, guinea pigs of the latter group exhibited higher levels of various acyl carnitines in their perilymph metabolome, suggesting an increased oxidative stress. Moreover, the noise only group showed lower levels of two osmoprotectans, namely stachydrine and homostachydrine, compared to the other three groups, indicating a potential osmolytic effect which may be attenuated by the H_2_ gas. 

A serious and unresolved clinical issue is ototoxicity due to treatment with cisplatin which is the most used anticancer drug [52]. In a study by Pierre et al. [47] on guinea pigs, changes in serum metabolomic profile induced by cisplatin treatment were investigated. Some animals were treated with sodium thiosulfate (STS), a candidate otoprotectant, 30 min before cisplatin administration; another group of animals received sodium chloride. Blood samples were collected before and 4 days after the treatment and metabolomics investigations were carried out through LC-MS. Cisplatin treatment induced a significant increase in hearing threshold and also a loss of outer hair cell in both groups. However, metabolomics analysis revealed that cisplatin had a major effect on serum metabolome, with only minor, non-significant differences observed in subjects treated with STS. Conversely, it was observed an inverse correlation between the concentration of four metabolites, namely N-acetylneuraminic acid, ceramides, cysteinylserine, L-acetylcarnitine, and the hearing threshold shift at high frequencies (20 and 30 kHz), but only in the group treated with sodium chloride. In particular, L-acetylcarnitine is a short-chain acylcarnitine which is involved in numerous functions, including proper mitochondrial function, removal of oxidative products, and regeneration of peripheral nerves [52,53].

## 4. Discussion

### 4.1. Main Findings 

In the realm of HL, a multifaceted and pervasive sensory impairment, lies a rich tapestry of research and discoveries. This intricate domain, encompassing the delicate mechanics of auditory function and the complexities of human communication, has captivated the scientific community for decades [10]. As we endeavour to explore its multifarious facets, we unravel the intricate threads of metabolomics, which offers a unique window into the mechanisms underlying the diverse phenotypes of HL. Although the emerging evidence in congenital and genetic inner ear diseases, little is known on major causes of HL. Moreover, in approximately 50% of post-lingual cases of SNHL, the site of the lesion and the molecular mechanisms involved remain unclear. In such a complex scenario the integration of metabolic profiling data related to most common condition including NIHL, SSNHL, Ménière’s disease, ototoxic drugs and ARHL is challenging to fully understand the pathophysiological mechanism, diagnose and treat these different phenotypes of HL [51].

Importantly, although the heterogeneous profiles under various disease conditions, inflammation and redox status dysregulation seem to characterize different combinations in the metabolomics pattern. Accordingly, the major experimental findings provide evidence on the interplay between redox status unbalance and inflammation as causative mechanisms targeting the hair cells, fibers toward the sensorineural epithelia and spiral ganglion neurons in models of cisplatin ototoxicity, NIHL and ARHL [52]. Hence, the emerging findings of metabolomics profiles in the inner ear tissues support the unbalance of glutamatergic synapses between the inner hair cells and spiral ganglion neurons undelaying the pathogenesis of NIHL and cisplatin ototoxicity [52,54]. The anatomical cochlear features, manly the bony capsule, the blood labyrinthine barrier and microscopic fine structure explain the low impact of disease-specific metabolites in the inner ear pathologies as compared to their expression in other conditions such as cancerogenesis, heart diseases or Alzheimer disease [55]. Thus, a limitation to reach a targeting approach to these pathologies remains the sample collection, the inaccessibility and fragility of the cochlea make very difficult the biopsy, imaging and fluid sampling of the cochlea. A promising source for collection of endolabyrinth fluids is represented by inner ear surgery techniques such as stapedectomy, labyrinthectomy, and cochlear implantation. If perilymph sampling during stapedectomy could be unsafe for hearing improving which is a mainstream of this surgery, inner ear fluids collected during labyrinthectomy or cochlear implantation procedure raise the possibility to targeting several condition such as Ménière disease and allow a comparison between congenital, genetic or acquired causes of HL respectively [56]. 

To date, a technical limitation of the inner ear fluids sampling is the risk of perilymph and endolymph mixture during invasive surgical procedures such as cochlear implantation which can compromise the metabolomics detection although the soft surgery techniques adopted for hearing preservation. Conversely, the urine and plasma/serum are potential sources for detectable metabolites, however the extremely low concentration and variability of circulating metabolites even at the baseline related to age, sex, ethnicity and others affect the perspective for a patient tailored approach to diagnosis and therapy [57]. As precision medicine is rapidly evolving, although some limitations it is likely that advances in the otology field will be achieved. 

In this review, we discuss key findings and implications of 13 human and 7 animal metabolomics studies focusing on various aspects of hearing HL.

In this context, the metabolomic profiling of the perilymph arouses considerable interest. Indeed, when studying localized biological effects, it is generally preferable to use tissues or samples taken close to the affected region, as they can be more representative of the system being studied. In this regard, the cochlear perilymph is in close contact with numerous structures involved in signal transduction mechanisms, including the auditory nerve, sensory hearing cells and spiral ganglion cell bodies [40]. Thus, since the late ‘60s researchers have been sampling this endolabirinthic fluid to analyze its composition, especially with regard to proteins [9,58,59]. In the case of SNHL, the investigation of the human perilymph metabolome has yielded interesting findings, revealing a compelling association between the duration of HL and the levels of N-acetylneuraminate [38], the most prevalent sialic acid in mammalian organisms. Given the pivotal role of sialic acids as fundamental constituents in the construction of glycan structures present in cell membrane glycoproteins and glycolipids, the authors posit that the heightened levels of *N*-acetylneuraminate may stem from the rupture of hair cell membranes. Verily, sialic acids hold great importance in a plethora of cellular functions, encompassing pivotal roles in cellular signalling, mechanisms of adhesion and migration, as well as implications in cancer metastasis and infectious processes. Alterations in their metabolic pathways have already been described in several pathological conditions [60,61]. 

However, while intriguing, these findings are constrained in their reliability due to the small sample size, and should be confirmed with further research. 

Overall, the analysis of human perilymph holds great potential for providing valuable information to develop novel mechanistic hypotheses. Nonetheless, the collection of these samples while safeguarding against the risk of iatrogenic HL is still a major challenge. Also, obtaining samples during surgical procedures entails a notable risk of intraoperative contamination between endolymph and perilymph, so it could be appropriate to indicate the endolabyrinth fluids [56]. Apart from these intricacies, a further technical impediment emerges in the analysis of exceedingly minute sample volumes, frequently falling within the sub-microliter spectrum. This challenge is rooted in the limited accessibility to perilymphatic fluid. Nonetheless, contemporary LC-MS technology has demonstrated its capability to surmount this constraint. In point of fact, Mavel et al. [62] have elucidated that notwithstanding a dilution ratio surpassing 1:100 of perilymphatic fluid with extraction and reconstitution solvents, the utilization of high-resolution LC-MS facilitates the robust identification and quantification of at least 100 metabolites.

On the other hand, a major advantage of metabolomics is that the entire metabolome is deeply influenced by the individual’s pathophysiological condition. As a result, a distinctive metabolomic signature of the pathology can be identified in several potential samples, while still offering valuable insights into the underlying biochemical processes associated with the condition [11]. The most interesting samples for human studies are those of urine and blood. Undoubtedly, these samples boast a wealth of information in terms of metabolomics, yet their collection merely necessitates non-invasive or minimally invasive procedures [63,64]. Metabolomic analyses of serum samples from patients with SSNHL have shed light on significant disturbances in fatty acid metabolism and its potential impact on hearing prognosis [42]. Indeed, the inner ear, being a highly energy-demanding organ, undergoes enhanced fatty acid oxidation when glucose metabolism falls short in meeting its energy requirements. These observations are also consistent with previous research describing the association of fatty acid metabolism with oxidative stress injury and apoptosis of inner hair cells, ultimately leading to acute hearing impairment [45,46]. In a study on ASNHL, urinary VOCs levels were investigated as indicators of oxidative stress, finding an inverse correlation between hearing threshold and mercapturic acids concentration [36]. Oxidative stress adversely affects the cochlea by generating reactive oxygen species that harm DNA, disrupt lipids, and trigger apoptosis within the cochlear tissues [65], thus emphasizing the necessity for additional studies to explore potential therapeutic implications. 

In addition, an intriguing topic that has captivated great attention in both human and animal studies is that of NIHL, a sensorineural deafness usually resulting from prolonged exposure of the auditory system to a noisy environment. This condition represents a significant public health issue, especially due to its high incidence among young individuals, as sources of high noise exposure often stem from occupational settings or recreational activities. Overall, sound intensities surpassing 85 dB are considered as potentially detrimental [66,67], and it is estimated that every day approximately 10% of the global population is subjected to sounds that can be damaging [68]. Nevertheless, little is known about the mechanisms underlying this damage. Studies on animals revealed several metabolic alterations induced by noise exposure, with Fujita et al. [46] focusing on the metabolomic profile of guinea pig inner fluids and Ji et al. [49] analyzing the entire mouse inner ear. The two studies, while exploring the same phenomenon, yielded inconsistent results. However, these disparities may be attributed to substantial methodological differences, rendering the findings essentially incomparable. Another study on mice revealed a remarkable 2.85 fold increase in spermidine levels in cochlea tissues taken from the noise-exposed group compared to controls [50]. Spermidine, that is a major inducer of autophagy, is associated with anti-inflammatory responses [69] and participates in metabolic pathways linked to β-alanine and arginine and proline, which were found to be significantly affected by noise exposure. Hence, it is reasonable to speculate that spermidine upregulation may play a role in the cochlear injury mechanism of NIHL through autophagy regulation. Additionally, alterations in the β-alanine pathway were also observed in the perilymph of sheep following noise exposure [51]. In humans, Miao et al. [39] and Zhang et al. [40] independently investigated NIHL-related metabolites in workers exposed to factory noises. The findings of the two studies overlapped with regard to retrograde endocannabinoid pathway and choline metabolism, which emerged as common biological processes, suggesting their significant involvement in NIHL development.

Endocannabinoids, in particular, have a great impact on the central nervous system, as they play a significant role in regulating synaptic function. Also, their signaling has been found to be crucial for the development, function, and survival of cochlear hair cells [70,71].

### 4.2. Future Perspectives

Metabolomics shows great promise as a powerful tool for delving into the intricacies of HL, a multifaceted topic with much yet to be explored. The main findings discussed so far represent promising steps towards enlightening the pathophysiological pathways associated with etiologies of SNHL. Nonetheless, the current state of the art is not without significant limitations. 

To date metabolomics analysis on plasma, urine, perilymph samples and in vitro cells derived from HL patients and animals has identified disease-specific metabolic pathways and metabolites. Most of the included metabolomic studies investigated NIHL [36,39,40,45,46,48,49,50,51] while other studies have investigated SSNHL [34,42] and HL related to aging [35,41] and ototoxic drugs [30,33,47]. Therefore, currently not all deafness endophenotypes are been equally characterized according to metabolic signature, for better understanding of the different pathways involved in HL. Additionally, forms of HL as SSNHL, NIHL and acoustic trauma represent acute form of HL that are presumed to be reversible and the clinical courses and therapeutic responses of those types of HL markedly differ. Therefore, is possible that the inflammatory and oxidative stress-related pathways involved in mediating cochlear damage during the acute stage and the metabolites identified do not provide the same diagnostic and predictive value for permanent HL. Overall, based to published data, metabolomics appears as a promising tool to define the etiology, duration such as acute and permanent/chronic condition and management of HL.

In NIHL the metabolomics analysis identified several previously unknown pathways [51] that were altered by noise exposure related to mechanical destruction, neurotransmission, damages of the synapses and nerve, and oxidative stress due to inflammatory reactions. This indicates that this omic technique may provide new insights from a metabolic perspective to explain the pathophysiological mechanisms of this disease. Moreover, the absence of standardized methods for analysis poses a challenge in assessing the reliability of the results obtained from different studies. Additionally, the variability in sample sizes and the diverse focus of the investigations further complicate drawing definitive conclusions. To fully unlock the potential of metabolomics in HL research, it is imperative for future investigations to address these limitations. Standardization of methodologies and larger, well-controlled studies will be essential to establish robust and reliable results and unveil comprehensive metabolic pathways associated with different types of HL. 

These limitations not only exert their influence on metabolomics investigations related to HL, but also extend to the broader spectrum of metabolomics applications within the medical domain. Consequently, a concerted endeavor is currently underway to establish collective quality control prerequisites and fundamental criteria for the dissemination of data and findings [72,73,74]. This endeavor assumes paramount importance, serving to enhance the congruity of acquired results and bolster the dependability thereof. In this regard, numerous studies chronicled in this review, by way of illustration, omit the provision of specifics concerning the extent of reliability attributed to the structural identification of the documented metabolites, as stipulated by the Metabolomics Standards Initiative (MSI) [75,76]. Evidently, this constitutes an indispensable stride in the realm of formulating hypotheses regarding potential pathological mechanisms underpinning observed conditions arising from distinct biochemical perturbations.

Likewise, the frequently encountered, inadequately substantiated divergence in pre-processing methodologies applied to the data, along with the strategies employed for data analysis, collectively engender a state of limited comparability among scrutinized outcomes. Addressing this facet also demands meticulous standardization endeavors. 

Moreover, most studies are based on a retrospective case-control experimental design. This, while providing a valid representation of disease state versus controls, does not allow disambiguation between cause-and-effect of the observed metabolomic changes. Therefore, future research should include preferably prospective studies that can contribute to this purpose, thus offering prospects for the development of novel and robust mechanistic hypotheses.

By embracing these advancements, metabolomics can emerge as a transformative force in shedding light on the complexities of hearing impairment and opening new avenues for diagnosis, treatment, and the development of personalized interventions in the field of audiology.

## 5. Conclusions

In conclusion, the application of metabolomics in HL research offers valuable insights into the underlying mechanisms and diverse phenotypes of this condition. Thus, metabolomic analysis will fill the gap on the understanding of biomarkers in biological fluids easier collected such as urine or serum and promote development of new therapies based on a personalized medicine approach. 

While existing studies have identified metabolic changes potentially associated with hearing impairment mainly in perilymph samples, which is still challenging for the limiting side effects. 

Future investigations with larger cohorts are needed to enhance reliability and foster novel mechanistic hypotheses. Standardization of methodologies is also essential to ensure consistency and comparability across studies. Embracing these advancements, metabolomics holds the potential to revolutionize our understanding of HL in its multifaceted aspects.

## Figures and Tables

**Figure 1 ijms-24-15188-f001:**
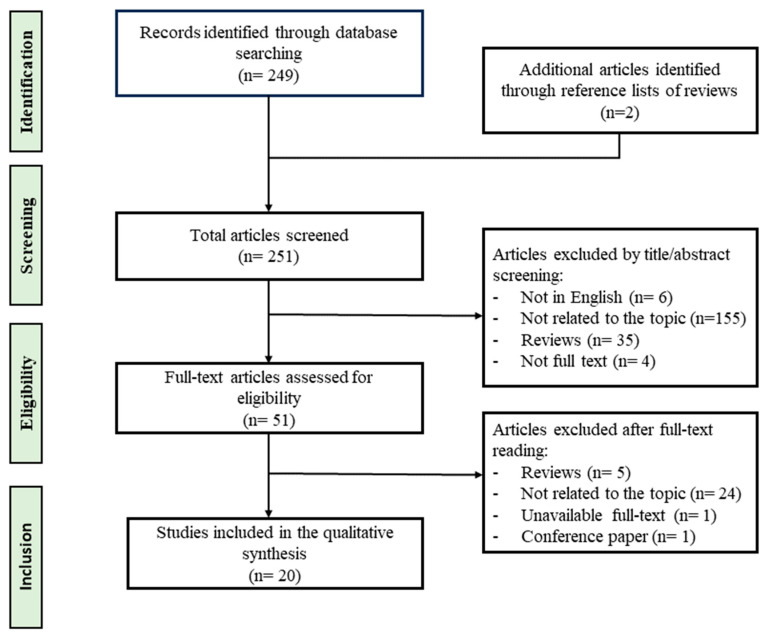
Flow diagram summarizing the study selection process.

**Table 1 ijms-24-15188-t001:** Metabolomic studies in human subjects with hearing loss. Abbreviations: AD: Alzheimer’s Disease; HC: Healthy Controls; HL: Hearing Loss; ISSNHL: Idiopathic Sudden Sensorineural Hearing Loss; NE: Noise Exposure; NIHL: Noise Induced Hearing Loss; PS: Presbycusis subjects; SSNHL: Sudden Sensorineural Hearing Loss.

Study/Country	Study Design	Subjects (Male/Female Ratio); Mean Age (years)	Samples	Metabolomic Analysis	Key Findings
Trinh et al., 2019 France [38]	Cross-sectional study	19 subjects (11 M, 8 F) 51 ± 28	Perilymph	LC-HRMS	A total of 106 metabolites were identified. The metabolomic profiles showed notable distinctions between individuals with HL lasting ≤ 12 years and those with HL exceeding 12 years. The discriminant compounds that stood out included N-acetylneuraminate, glutaric acid, cystine, 2-methylpropanoate, butanoate, and xanthine. N-acetylneuraminate levels were associated with the duration of HL.
Xu et al., 2020 USA [30]	Cross-sectional study	720 subjects (395 M, 325 F) 15.41 ± 2.22	Urine	UPLC-MS/MS	Significant positive association between pyrethroid exposure and hearing loss in US adolescents, estimated by measuring urinary levels of 3-BPA.
Long &Tang, 2021 USA [32]	Retrospective cross-sectional study	849 subjects (422 M, 427 F) 43.06 ± 14.04	Urine	UHPLC-ESI-MS/MS	No urinary metabolites of caffeine have been associated with changes in hearing threshold in U.S. adults.
Li et. al., 2022 USA [31]	Observational cross-sectional study	5537 subjects (2781 M, 2756 F) 4200 Adults Mean age: 42.08 ± 0.43 y 1337 adolescents Mean age: 15.32 ± 0.06 y	Urine	HPLC/MS/MS	Positive associations emerged between some urinary metabolites of PAHs (3-OHFlu, 2-OHFlu, 2&3-OHPh) and hearing loss, at total, speech and high frequencies in both adults and adolescents. The associations were similar among smokers and slightly different among non-smokers. Among adolescents, associations were also found between some urinary metabolites of PAHs and systemic inflammation, but this did not seem to influence hearing loss.
Llano et al., 2020 USA [41]	Observational study	185 subjects (95 M, 90 F): 40 AD + HL (27 M, 13 F) 77.2 ± 5.8 145 AD (68 M, 77 F) 74.8 ± 7.7	Serum	LC-MS	A total of 349 serum lipid metabolites were measured. Significant reduction in phosphatidylcholines levels was observed in AD patients with HL.
Miao et al., 2021 China [39]	Pilot study	124 subjects (All M): 62 NIHL Mean age: 42.11 ± 10.15 62 HC Mean age: 41.73 ± 10.07 y	Plasma	UHPLC-Q-TOF MS	59 metabolites among the 200 detected were found deeply altered in NIHL. Organic acids, including homodeoxycholic acid and quinolacetic acid, along with 3,4-dihydroxymandelic acid were significantly up-regulated. While most of downregulated were lipids, such as phosphatidylcholine, phosphatidylethanolamine, phosphatidylinositol.
Chou et al., 2020 USA [33]	Cross-sectional study	1071 participants (494 M, 577 F) 42.50 ± 22.00	Urine	GC-HRMS	The study reported a positive association between PAH metabolites and an increase in log-transformed hearing threshold by high- and low-frequency pure-tone audiometry. 1-Naphthol and 2-naphthol were the PAH metabolites most commonly found in urine. Moreover, a dose-response relationship was observed between PAH exposure and the increase in hearing thresholds.
Carta et al., 2017 Italy [34]	Prospective study	26 subjects (12 M, 14 F): 21 ISSNHL patients (10 M, 11 F) 47.6 (range of 9–73 years) 5 HC (2 M, 3 F) 41.4 (range of 23–61 years)	Urine	NMR	The authors identified noteworthy variations between individuals who positively responded to steroid therapy and those who did not. Those who didn’t respond showed elevated levels of β-alanine, trimethylamine N-oxide (TMAO), and 3-hydroxybutyrate, while citrate and creatinine levels exhibited an opposite trend in responders.
Dong et al., 2013 China [35]	Case-control study	20 subjects (11 M, 9 F): 11 PS (6 M, 5 F) 66.5 ± 2.4 9 HC (5 M, 4 F) 65.0 ± 1.8	Urine	GC-TOF/MS	Significant differences in urine metabolomic profiles were evident between the presbycusis patients and the control group. The analysis revealed 23 differentially expressed metabolites, closely associated with glutathione metabolism, amino acid metabolism, glucose metabolism, the N-methyl-d-aspartic acid (NMDA) receptor pathway, and the γ-aminobutyric acid (GABA) receptor pathway.
Pudrith & Dudley, 2019 USA [36]	Retrospective study	849 subjects (424 M, 425 F) 292 NE (71% M, 29% F): 45.81 y 557 non-NE (39% M, 61% F): 42.9 y	Urine	UPLC-MS/MS	Upon accounting for age, the high-frequency pure-tone thresholds exhibited a noteworthy increase of approximately 3 to 4 dB in the quartile groups with higher concentrations of five specific metabolites. These metabolites were identified as glutathione-dependent mercapturic acids and originated from parent compounds like acrylonitrile, 1,3-butadiene, styrene, acrylamide, and N,N-dimethylformamide. These associations were only reported in individual without noise exposure.
Zhang et al., 2022 China [40]	Case-control study	90 subjects (All M): 60 NE subjects (n = 30 with NIHL, n = 30 without NIHL): 39.28 ± 7.32 y 30 HC: 38.96 ± 7.32 y	Plasma	HPLC-MS/MS	A total of 6 pathways were found to be linked to both noise-induced hearing loss (NIHL) and noise exposure. These pathways include retrograde endocannabinoid signaling, sphingolipid signaling pathway, vitamin digestion and absorption, Fc gamma R-mediated phagocytosis, phospholipase D signaling pathway, and central carbon metabolism in cancer. 7 metabolites emerged as crucial in distinguishing NIHL patients from the other two groups. Among them, Pro-Trp, adenine, and dimethylglycine showed a gradual decrease, with the lowest levels observed in the NIHL group.
Wang et al., 2022 China [42]	Longitudinal study	40 subjects: 20 SSNHL patients (10 M, 10 F): 41.5 (range of 30.0–44.8 y) 20 HC (12 M, 8 F) 37.0 (range of 32.3–42.8 y)	Serum	LC/MS	Serum metabolomic analysis identified distinctive metabolite signatures in SSNHL patients compared to HC, with a significant involvement of fatty acids and sphingolipids metabolism. Serum levels of specific metabolites, including N4-acetylcytidine, sphingosine, and nonadecanoic acid, showed good predictabilities for hearing recovery in SSNHL patients.
Long & Tang 2022 China [37]	Cross-sectional study	726 subjects (341 M, 385 F); 42.81 ± 13.62 y (range of 20–69 y)	Urine	LC-ESI-MS	Positive association between 3-PBA levels and low-frequency and high-frequency hearing thresholds in participants aged 20–39 years.

**Table 2 ijms-24-15188-t002:** Metabolomic studies on animal models of hearing loss.

Study/Country	Animal Model	Hearing Loss	Samples	Metabolomic Analysis	Key Findings
Ji et al., 2019 USA [49]	Mice	Noise induced	Inner ear tissues	LC-MS/MS	A total of 40 metabolites exhibited differential responses to noise exposure. Among these, 25 metabolites were up-regulated, encompassing nucleotides, cofactors, carbohydrates, and glutamate, while 15 metabolites were down-regulated, primarily comprising amino acids. Moreover, the impact of noise on the inner ear metabolome in mice was found to be contingent on both the intensity and duration of exposure.
He et al., 2017 New Zealand China [48]	Rats	Noise induced	Brain tissue	GC-MS	88 metabolites were identified from the analysis of 12 different brain regions. The levels of 17 metabolites were found to be significantly affected by acoustic trauma in at least one of the analyzed brain areas. These were mostly involving amino acid metabolism, i.e., alanine, aspartate, glutamate, arginine, proline, and purine metabolic pathways, as well as urea cycle and oxidative reactions.
Pirttilä et al., 2019 USA [45]	Guinea pigs	Noise induced	Perilymph	HILIC-UHPLC-Q-TOF–MS	Guinea pigs exposed to noise and treated with Hydrogen gas (H2) showed reduced hearing loss compared to those exposed to noise alone. Metabolomic analysis confirmed the otoprotective role of H2 by revealing a closer similarity in the perilymphatic metabolome of H2-treated animals exposed to noise to that of control and H2 only groups, compared to the noise only group. Additionally, the noise only group exhibited higher levels of various acyl carnitines, while lower levels of osmoprotectans stachydrine and homostachydrine were observed.
Fujita et al., 2015 Japan [46]	Guinea pigs	Noise induced	Inner ear fluid, Plasma	GC-MS	Higher levels for 6 metabolites (ascorbic acid, fructose, galactosamine, inositol, pyruvate + oxaloacetic acid and meso-erythritol) and lower levels for 9 (phosphate, valine, glycine, glycerol, ornithine, glucose, citric acid + isocitric acid, mannose and trans-4-hydroxy-l-proline) were observed in the inner ear fluid than in plasma. The levels of 10 metabolites (3-hydroxy-butyrate, glycerol, fumaric acid, galactosamine, pyruvate + oxaloacetic acid, phosphate, meso-erythritol, citric acid + isocitric acid, mannose and inositol), changed significantly in the inner ear fluids as a consequence of noise exposure.
Pierre et al., 2017 Sweden [47]	Guinea pigs	Chemical induced	Serum	LC-MS	Some animals received sodium thiosulfate (STS), a potential otoprotectant, 30 min before cisplatin administration, while another group received sodium chloride. Non-significant differences were observed between the two groups. On the other hand, 4 metabolite changes showed significant correlation with high-frequency hearing loss, but only in the group receiving NaCl. These metabolites were N-acetylneuraminic acid, L-acetylcarnitine, ceramides, and cysteinylserine. Notably, a higher increase in the level of metabolite change was associated with a lower hearing threshold shift.
Miao et al., 2022 China [50]	Mice	Noise induced	Cochlea tissue	GC-MS	Exposure to noise caused alterations in the levels of 17 metabolites, of which only 3 (spermidine, 3-hydroxybutyric acid and orotic acid) had increased levels. In addition, 9 metabolic pathways, namely β-alanine metabolism, pyrimidine metabolism, cAMP signaling pathway, butanoate metabolism, ketone body synthesis and degradation, arginine and proline metabolism, GABAergic synapse, insulin secretion, and purine metabolism, were significantly affected by noise exposure.
Boullaud et al., 2022 [51]	Sheep	Noise induced	Perilymph	LC-MS	Following noise exposure, certain metabolites demonstrated an increasing trend, including urocanate, oleate, 5-Oxo-L-Proline, N-Acetyl-Glucose, N-Acetylneuraminate, L-Tyrosine, Trigonelline, Leukotriene-B4, 5,6-Dihydrouracil, and 3-Ureidopropionate. Conversely, a decreasing trend was observed for deoxycarnitine, L-Carnitine, N-Acetyl-L-Leucine, S-(5′-Adenosyl)-L-Homocysteine, and epinephrine. In addition, 5 metabolic pathways were found to be involved: phenylalanine/tyrosine/tryptophan metabolism, β-alanine metabolism, pantothenate and CoA biosynthesis, pyrimidine metabolism, and amino and nucleotide sugar metabolism.

## Data Availability

No new data were created or analyzed in this study. Data sharing is not applicable to this article.

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
