# Peer review of "A Systematic Review on Metabolomics Analysis in Hearing Impairment: Is It a Possible Tool in Understanding Auditory Pathologies?"

_ijms, 2023, doi:10.3390/ijms242015188_

Round 1
Reviewer 1 Report
This review describes metabolomic changes possibly related to hearing loss and ototoxicity in humans and animal models. The authors analyzed findings from 20 research articles and listed changes of metabolites in 2 tables. They also present some challenges that remain to be addressed in future works. This topic is potentially interesting but the manuscript needs significant improvements to provide mechanistic insights for understanding the etiology of hearing impairment.
1. The review is very descriptive and superficial without an in-depth and critical analysis of possible relationships between metabolomic changes and hearing loss. It is unclear whether changes of metabolites represent a direct or indirect consequence of auditory pathologies or whether these changes are responsible for hearing loss. The authors should make efforts to further discuss how the identified compounds may be related to the dysfunction of inner ear or hair cells and how they may be served as therapeutic targets. With a relative lack of mechanistic insights, the review does not bring the gap in understanding auditory pathologies.
2. In table 1, the authors listed metabolomic studies in human subjects with hearing loss, but for many subjects, the types of hearing loss were not described. In addition, it will be interesting to indicate whether there are metabolomic studies in subjects with mutations or dysfunctions of deafness genes.
3. It will help the readers and make the discussion clear if the sections 3.1 and 3.2 are subdivided using sub-headings, according to the types of auditory pathologies or the types of compounds identified, or other criteria.
4. For a review, the titles of sections 2 and 3 are not appropriate.
Author Response
This review describes metabolomic changes possibly related to hearing loss and ototoxicity in humans and animal models. The authors analyzed findings from 20 research articles and listed changes of metabolites in 2 tables. They also present some challenges that remain to be addressed in future works. This topic is potentially interesting but the manuscript needs significant improvements to provide mechanistic insights for understanding the etiology of hearing impairment.
We thank for positive overall comment; however, we respectfully express our differing viewpoint. The aim of a systematic literature review is not the release of new mechanistic hypotheses. There are specific editorial formats for this type of evaluation. In any case currently it is good practice to support hypotheses with scientific and not speculative evidence, and this should be done using the original article tool and not the literature review. Specifically, the role of reviews in the medical field is: "… to identify, evaluate, and summarize the findings of all relevant individual studies over a health-related issue, thereby making the available evidence more accessible to decision makers." as reported only as one of thousands of examples by " Gopalakrishnan S, Ganeshkumar P. Systematic Reviews and Meta-analysis: Understanding the Best Evidence in Primary Healthcare. J Family Med Prim Care. 2013 Jan;2(1):9-14. doi: 10.4103/2249-4863.109934. PMID: 24479036; PMCID: PMC3894019."
- The review is very descriptive and superficial without an in-depth and critical analysis of possible relationships between metabolomic changes and hearing loss. It is unclear whether changes of metabolites represent a direct or indirect consequence of auditory pathologies or whether these changes are responsible for hearing loss. The authors should make efforts to further discuss how the identified compounds may be related to the dysfunction of inner ear or hair cells and how they may be served as therapeutic targets. With a relative lack of mechanistic insights, the review does not bring the gap in understanding auditory pathologies.
Accordingly, to your suggestion new comments have been added in the Discussion section. The title has been changed to meet the approval of the Reviewer considering that the metabolomics could be a tool in understanding auditory pathologies. However, disambiguation between cause and effect is a sensitive topic in scientific research and requires specific experimental designs, generally prospective. Case-control studies of the retrospective type representing the instantaneous condition generally do not allow this insight.
In the light of the reviewer’s comment, we have updated the 'Future Perspectives' section to acknowledge the need for prospective studies. However, attempting to extend our review beyond the existing literature would be speculative and beyond the intended focus.
While we acknowledge the potential importance of discussing mechanistic insights and therapeutic avenues, we aimed to provide a comprehensive analysis of the available data.
We believe our review effectively summarizes the metabolomic changes associated with hearing loss based on the available evidence.
- In table 1, the authors listed metabolomic studies in human subjects with hearing loss, but for many subjects, the types of hearing loss were not described. In addition, it will be interesting to indicate whether there are metabolomic studies in subjects with mutations or dysfunctions of deafness genes.
In Table 1 we have detailed the etiology of hearing loss whenever this was present in the original article. In cases where the original article does not give specific details, we have referred to patients only as HL, consistent with published evidence. While we concur with the reviewer's suggestion regarding delving into the metabolic alterations arising from specific gene mutations linked to hearing impairment, we must emphasize once again that a systematic review is inherently limited to summarizing existing data. The process does not involve the generation of novel information. Regrettably, given the absence of such inquiries within the included papers, it remains unfeasible to incorporate these results into our manuscript.
- It will help the readers and make the discussion clear if the sections 3.1 and 3.2 are subdivided using sub-headings, according to the types of auditory pathologies or the types of compounds identified, or other criteria.
We thank the reviewer for the suggestion. We have divided sections 3.1 and 3.2 into sub-sections according to the type of sample analysed. In particular, section 3.1, which concerns metabolomics studies on humans, has been divided into three sections, relating to studies on perilymph, on samples derived from blood and on urine. Section 3.2, concerning studies on animals, in light of the small number of studies in the literature, has been divided into only two sub-sections, namely tissues and biofluids.
- For a review, the titles of sections 2 and 3 are not appropriate.
We thank the reviewer for the feedback. However, the titles of sections 2 and 3, i.e. 'Materials and Methods' and 'Results and Discussion', are part of the journal template and it is not up to the authors to change them. Personally, we feel it is more than fair on the part of the journal editors to include these sections as such. Indeed, even in literature reviews, it is appropriate to clearly provide details about the literature search methodology and the results derived from it in a rigorous manner.
In any case, we feel that discussions about the journal's template are not related to our article, but the reviewer may contact the journal's editorial office to express his or her displeasure about the presence of the “Materials and Methods” and “Results and Discussion” sections in the reviews and accordingly we are able to change the titles.
Reviewer 2 Report
This systematic review addresses a critical issue in hearing research by juxtaposing current knowledge of the inner ear and other body fluids metabolome concerning hearing disorders. A laudable intention and carefully executed - the manuscript is of real value and enjoyable to read.
I only have a few comments:
„As the metabolome lies downstream from the central dogma of biology” – what precisely is meant here? “Dogma of biology” – how do you define this term? I suggest revising this sentence (e.g., “The metabolome reflects both the genetic characteristics of an organism and the environmental influences”).
Line 126: “with N-acetylneuraminate showing a strong correlation with the duration of HL” – please indicate if it was a positive or negative correlation.
Lines 133-134: “with significant associations between serum N4-Acetylcytidine, sphingosine, and nonadecanoic acid levels with hearing recovery in SSNHL patients” – again here, please indicate if the correlations were positive or negative.
In Table 1, sometimes the authors use an arrow (“4200 Adults → Mean”) and sometimes not. What is the meaning of these arrows? Are they necessary? I find them confusing.
Line 155 “KEGG pathway“ please define.
Line 158 “accumulation of kidney deficiency points” What is a kidney deficiency point? Please define.
Line 208, please spell out the “HC”.
Lines 217 – 221 – this belongs to the paragraph below (animal studies). Please move.
In the study of Fujita et al., 2015 [46], a sample analyzed was the “inner ear fluid”. Please go back to that publication, explain precisely what that meant, and provide that information.
Finally, I am missing a paragraph in which the authors would shortly explain the technologies used in the studies (perhaps in a tabular form: LC-MS, LC-MS/MS, GC-MS, HILICUHPLC-QTOF–MS, LC-ESI-MS, HPLC-MS/MS, UPLC-MS/MS, GC-TOF/MS, GC-HRMS, UHPLC-QTOF MS, LC-HRMS, and NMR), provide their full names and underline the advantages and disadvantages for the use in hearing research (e.g., sample type, sample size, accuracy, sensitivity, capacity and type of metabolites that can be detected).
Author Response
This systematic review addresses a critical issue in hearing research by juxtaposing current knowledge of the inner ear and other body fluids metabolome concerning hearing disorders. A laudable intention and carefully executed - the manuscript is of real value and enjoyable to read.
I only have a few comments:
„As the metabolome lies downstream from the central dogma of biology” – what precisely is meant here? “Dogma of biology” – how do you define this term? I suggest revising this sentence (e.g., “The metabolome reflects both the genetic characteristics of an organism and the environmental influences”).
We extend our gratitude to the reviewer for their insightful comment. It has come to our attention that an important adjective, "molecular," was inadvertently omitted before the term "biology." We have duly rectified this oversight. The elucidation of the central dogma of molecular biology, attributed to Crick in 1958, remains a pivotal framework. This paradigm posits a unidirectional flow of information within a cell—commencing from DNA to RNA, onward to proteins, and culminating in metabolites. While exceptions have surfaced over time, this fundamental construct remains a cornerstone of molecular biology, casting light upon a myriad of intricate biological phenomena.
Line 126: “with N-acetylneuraminate showing a strong correlation with the duration of HL” – please indicate if it was a positive or negative correlation.
We thank the reviewer for the comment. We included the verse of the correlation.
Lines 133-134: “with significant associations between serum N4-Acetylcytidine, sphingosine, and nonadecanoic acid levels with hearing recovery in SSNHL patients” – again here, please indicate if the correlations were positive or negative.
We thank the reviewer for the comment. We included the verse of the correlation.
In Table 1, sometimes the authors use an arrow (“4200 Adults → Mean”) and sometimes not. What is the meaning of these arrows? Are they necessary? I find them confusing.
As suggested by the reviewer, we made the tables clearer by removing all the arrows.
Line 155 “KEGG pathway“ please define.
KEGG (Kyoto Encyclopedia of Genes and Genomes) is a collection of databases dealing with genomes, biological pathways, diseases, drugs, and chemical substances. KEGG is utilized for bioinformatics research and education, including data analysis in genomics, metagenomics, metabolomics and other omics studies, modeling and simulation in systems biology, and translational research in drug development. As suggested, we defined the acronym in the main text.
Line 158 “accumulation of kidney deficiency points” What is a kidney deficiency point? Please define.
As suggested, we defined the “kidney deficiency scoring” according to the author description in the original paper. It is a scoring A scoring system for kidney deficiency symptoms generated according to “Standards of practice for evaluating the grades of kidney deficiency syndrome differentiation factors” by “Standards of reference of TCM deficiency syndrome” revised by the Chinese Integrated Traditional and Western Medicine Deficiency Syndrome and Geriatrics Research Professional Committee
Line 208, please spell out the “HC”.
In the current manuscript version, we declared the HC acronym at its first occurrence, please see line 134-135 (beginning of the section 3.1.2)
Lines 217 – 221 – this belongs to the paragraph below (animal studies). Please move.
We hold the reviewer's comment in high regard. Nevertheless, it's worth noting that the investigation by Zhang et al. pertained to human subjects, encompassing a cohort of 60 individuals exposed to noise. Half of this group experienced noise-induced hearing loss (NIHL), while the other 30 participants served as healthy controls (HC). Of particular interest, Zhang et al. fortify their findings by delving into an in-silico gene expression assessment, skillfully leveraging the publicly accessible dataset GSE8342. Given the primary focus on human-based empirical research, we aptly situated this study within the human-centered segment of our work, encompassing the comprehensive spectrum of outcomes reported therein.
In the study of Fujita et al., 2015 [46], a sample analyzed was the “inner ear fluid”. Please go back to that publication, explain precisely what that meant, and provide that information.
Throughout the entirety of their manuscript, as well as within the title itself, the authors consistently alluded to the term "inner ear fluid." It is with this consistent intention in mind that we maintained this particular definition. Nevertheless, in response to the reviewer's valuable suggestion, we have taken steps to elucidate that the author's use of this term was intended to encompass the amalgamation of both endolymphatic and perilymphatic fluid components.
Finally, I am missing a paragraph in which the authors would shortly explain the technologies used in the studies (perhaps in a tabular form: LC-MS, LC-MS/MS, GC-MS, HILICUHPLC-QTOF–MS, LC-ESI-MS, HPLC-MS/MS, UPLC-MS/MS, GC-TOF/MS, GC-HRMS, UHPLC-QTOF MS, LC-HRMS, and NMR), provide their full names and underline the advantages and disadvantages for the use in hearing research (e.g., sample type, sample size, accuracy, sensitivity, capacity and type of metabolites that can be detected).
The technologies mentioned in the review are in common use in metabolomics and a full description is beyond the scope of this paper. In any case, accepting the reviewer's suggestion, we have prepared a supplementary table summarizing the main advantages and disadvantages of all the technologies used in the reviewed papers. Table S1 is seen for the purpose.
Reviewer 3 Report
The authors searched Pubmed for articles on metabolomics and hearing loss and in the end found 13 human and 7 animal studies. The findings are described quite in detail and therefore dispersed, indicating that prospective studies of sufficient sizes are necessary to gain a better understanding of the reasons behind human hearing loss in the general population than that achieved from specialized studies of eg hearing liss and cisplatin metabolic effects in cancer patients.
I think the authors can add some insight to the readers by commenting why the review papers could not add anything to the text, and also discuss if acute hearing loss (which is often reversible) has got to much attention in the paper. After all, the permanent long term hearing loss is what we mainly should work on.
The title of the paper is too positive - the method is a possible way forward in understanding hearing loss, unproven so far.
Author Response
The authors searched Pubmed for articles on metabolomics and hearing loss and in the end found 13 human and 7 animal studies. The findings are described quite in detail and therefore dispersed, indicating that prospective studies of sufficient sizes are necessary to gain a better understanding of the reasons behind human hearing loss in the general population than that achieved from specialized studies of eg hearing liss and cisplatin metabolic effects in cancer patients.
I think the authors can add some insight to the readers by commenting why the review papers could not add anything to the text, and also discuss if acute hearing loss (which is often reversible) has got to much attention in the paper. After all, the permanent long term hearing loss is what we mainly should work on.
The title of the paper is too positive - the method is a possible way forward in understanding hearing loss, unproven so far.
We thank the reviewer for the overall positive comment. As suggested we have added new comments in the 'Future Perspectives' section to discuss the need to better characterize all hearing loss endophenotypes according to metabolic signature and especially permanent hearing loss.
Accodingly to the Reviewer’s suggestion the title has been changed.
Round 2
Reviewer 1 Report
Sections 2 and 3 are not divided into subsections. It is difficult to find the information.
I cannot find the discussion section with new comments added. The authors should reorganize the manuscript with a discussion section.
Author Response
We thank the reviewer for the feedback. In accordance with the reviewer's suggestion, we have revised the subdivision of the manuscript. We have subdivided the "Results and Discussion" sections, now paragraphs 3 and 4, respectively, which are in turn subdivided into sub-sections.
Specifically, paragraph 3 "Results" (line 113) is subdivided into:
3.1. Metabolomic profiling in humans with hearing loss (line 114), of which 3.1.1 Metabolomic analysis on perylimph samples (line 126), 3.1.2 Metabolomic analysis on blood-derived samples (line 133), 3.1.3 Metabolomic analysis on urine samples (line 178)
and
3.2 Metabolomics investigations on animal models of hearing loss (line 238)
of which
3.1.2 Metabolomic analysis on tissues (line 254)
3.1.2 Metabolomic analysis on biofluids (line 266).
Section 4 "Discussion" (line 321) is in turn divided into 4.1 Main Findings (line 322) and 4.2 Future perspectives (line 453).
We consider the paper to be well divided and do not believe that further subdivisions would add value to the manuscript. In particular, we feel it would be excessive to subdivide paragraph 2 (Methods) into sub-sections, as it is only 10 lines on literature search methods (lines 99-109).
Round 3
Reviewer 1 Report
No further comments.